# Therapeutic Potential of Natural Products in the Treatment of *Schistosomiasis*

**DOI:** 10.3390/molecules28196807

**Published:** 2023-09-26

**Authors:** Carine Machado Azevedo, Cássio Santana Meira, Jaqueline Wang da Silva, Danielle Maria Nascimento Moura, Sheilla Andrade de Oliveira, Cícero Jádson da Costa, Emanuelle de Souza Santos, Milena Botelho Pereira Soares

**Affiliations:** 1Gonçalo Moniz Institute, Oswaldo Cruz Foundation (IGM-FIOCRUZ/BA), Salvador 40296-710, Brazil; carine.cardoso@fiocruz.br (C.M.A.); cassio.meira@fieb.org.br (C.S.M.); 2SENAI Institute of Innovation in Health Advanced Systems (CIMATEC ISI SAS), University Center SENAI/CIMATEC, Salvador 41650-010, Brazil; jaquelinewang.sw@gmail.com (J.W.d.S.); emanuelle.santos@fieb.org.br (E.d.S.S.); 3Aggeu Magalhães Institute, Oswaldo Cruz Foundation (IAM-FIOCRUZ/PE), Recife 50740-465, Brazil; danielle.moura@fiocruz.br (D.M.N.M.); sheilla.andrade@fiocruz.br (S.A.d.O.); jadsoncosta02@gmail.com (C.J.d.C.)

**Keywords:** schistosomiasis, natural compounds, essential oil, molecular tools

## Abstract

It is estimated that 250 million people worldwide are affected by schistosomiasis. Disease transmission is related to the poor sanitation and hygiene habits that affect residents of impoverished regions in tropical and subtropical countries. The main species responsible for causing disease in humans are *Schistosoma Mansoni*, *S. japonicum*, and *S. haematobium*, each with different geographic distributions. Praziquantel is the drug predominantly used to treat this disease, which offers low effectiveness against immature and juvenile parasite forms. In addition, reports of drug resistance prompt the development of novel therapeutic approaches. Natural products represent an important source of new compounds, especially those obtained from plant sources. This review compiles data from several in vitro and in vivo studies evaluating various compounds and essential oils derived from plants with cercaricidal and molluscicidal activities against both juvenile and adult forms of the parasite. Finally, this review provides an important discussion on recent advances in molecular and computational tools deemed fundamental for more rapid and effective screening of new compounds, allowing for the optimization of time and resources.

## 1. Introduction

Schistosomiasis, a neglected tropical disease (NTD), infects around 250 million people worldwide [1,2]. Like other NTDs, it predominantly affects the poorer populations living in tropical and subtropical regions [2,3]. There are three main species of trematodes of the genus Schistosoma, the causative agents of the disease in humans: *S. mansoni* and *S. japonicum* (which cause intestinal and hepatosplenic disease) and *S. haematobium* (which affects the genitourinary system) [4]. The distribution of each species follows that of its respective intermediate host snail. In Africa and the Middle East, *S. haematobium* and *S. mansoni* are found, while the latter is also present in South America and the West Indies. However, *S. japonicum* is endemic to some Asian countries, such as China, the Philippines, Indonesia, and the Mekong Delta region [3]. Although other species, such as *S. intercalatum*, *S. guineensis*, and *S. mekongi*, are also known to cause the disease in humans, their importance is secondary since distribution is restricted to Central and West Africa (the first two species) and Laos and eastern Cambodia [2].

Infection occurs when an individual comes into contact with water containing cercariae released by intermediate host snails, which may be of the genus *Biomphalaria* (for *S. mansoni*), *Oncomelania* (for *S. japonicum*), or *Bulinus* (for *S. haematobium*). While cercariae can survive in water for 1 to 3 days [5], the ability to infect reduces rapidly within a few hours of release [6]. After penetrating the human host’s skin, the parasites circulate, reaching the lungs and then the liver where they transform into young worms or schistosomula. Within 4 to 6 weeks, the worms mature within the portal venous system, mate, and migrate in pairs to the mesenteric veins [5], in the case of *S. mansoni* and *S. japonicum*, or to the vesical venous plexus of the urogenital system, in the case of *S. haematobium* [3].

After mating, females begin to lay eggs that may be retained in the liver (forming granulomas and subsequent fibrosis) or pass through the intestine, returning to the environment along with feces. In the case of *S. haematobium*, eggs are excreted through the urine. The lifespan of adult worms generally ranges from 3 to 5 years, but reports have indicated survival for up to 30 years [5]. Eggs, upon coming into contact with water, hatch and release the miracidia that infect snails, within which asexual replication leads to the formation of the mother and offspring sporocysts, in addition to subsequent cercariae [5,7]. Around 28 to 30 days (*S. mansoni* and *S. haematobium*) or 90 days (*S. japonicum*) following infection, the snail begins to shed cercariae [3]. This process is stimulated by light, mainly occurring during the day [5].

Since infection occurs through contact with water contaminated by cercariae, poor sanitation and hygiene can place children, adolescents, and adults at risk [8]. Contamination occurs mainly in poor regions of developing countries, in which the prevalence of disease is higher [3]. Other activities involving contact with water, i.e., domestic (washing clothes and dishes in open bodies of freshwater), recreational (bathing in rivers and lakes), or professional activities can expose people to risk [8]. In this context, children and pregnant women are those most susceptible to infection and reinfection. In children, schistosomiasis can cause malnutrition, reduced growth, and cognitive impairment, while in pregnant women it can lead to premature birth, low birth weight, and higher maternal morbidity and mortality [9].

The pathology of schistosomiasis is closely related to the large quantity of eggs deposited by females, which accumulate in the liver/intestine or bladder/urogenital system [2,10]. However, studies have shown that host genetic factors may also be associated with more severe manifestations of disease (such as HLA class I and class II antigens) or with protection against severe hepatic fibrosis (such as HLA-DP alleles). In addition, the development of resistance to reinfection with *S. mansoni* has been linked to the SM1 locus located on chromosome 5q31–q33 [11]. The disease is characterized by three distinct phases: acute (following primary infection, more common in travelers or immigrants to endemic areas), established, and late chronic (commonly observed in individuals living in endemic areas) [2]. The acute phase (also known as Katayama fever or Katayama syndrome) commonly occurs in persons infected for the first time, with the symptoms exhibited in response to antigens released by schistosomula during the migration process, as well as by recently laid eggs [2,8]. This phase occurs between 2 weeks to 3 months after exposure to cercariae, with a typical clinical presentation consisting of fever, myalgia, headache, bloody diarrhea, hepatosplenomegaly, eosinophilia, non-productive cough, patchy infiltrates on chest radiography, and elevated IgE levels [2,3,5,10]. Residents in endemic areas do not exhibit an acute symptomatic phase, and usually develop an established active infection characterized by the presence of adult worms that produce eggs that are excreted in feces or urine. Although the adult worms present in blood vessels do not provoke an inflammatory reaction, eggs produce soluble antigens that induce the formation of granulomas that lead to tissue fibrosis. In the chronic phase, the accumulation of numerous granulomas provokes the formation of periportal fibrosis in the liver (*S. mansoni* and *S. japonicum*), leading to portal hypertension and the appearance of esophageal and gastric varices. These varicose veins can rupture and cause bleeding, potentially resulting in consequent death [3]. In the case of *S. haematobium*, patients present obstructive disease in the urinary and reproductive system, in addition to bladder calcification, genital lesions, kidney involvement, i.e., hydronephrosis and renal failure, and bladder cancer [3].

As neglected diseases, such as schistosomiasis, are not prioritized by pharmaceutical companies, therapeutic options are limited, outdated, and sometimes even non-existent. Despite being a disease that has been studied for several decades, the only effective treatment option against different species of *Schistosoma* is praziquantel (PZQ), which was discovered in 1972 and has been available for treatment since the 1980s [12]. In accordance with WHO recommendations, preventive chemotherapy involving PZQ is indicated for affected populations and at-risk groups. The frequency of treatment is determined by the prevalence of infection in school-aged children [13]. While the anthelmintic activity of PZQ remains uncertain, some authors speculate that it likely inhibits the Na^+^ and K^+^ pump in adult worms, increasing the permeability of the helminth membrane to certain monovalent and bivalent cations, such as calcium, which then leads to the intensification of muscle activity, followed by contraction and spastic paralysis [14,15]. While this drug achieves a cure rate of between 60 and 90%, it lacks activity against immature and juvenile parasites, which have been shown to survive drug exposure [3]. Consequently, new adult worms may appear 1 to 2 months after treatment, since administration consists of a single dose [2]. Other disadvantages involving the use of PZQ are related to its racemic nature, as only half of the dose is pharmacologically utilized and what is absorbed becomes rapidly metabolized into inactive metabolites, resulting in relatively minimal drug contact with parasites in the host’s bloodstream [16,17]. Despite advantages of low cost and facile administration in adults (single oral dose of 40 mg/kg) [9], there is no pediatric formulation for PZQ, the pills are large, and a bitter taste may further contribute to the low cure rates observed among preschool children [9,10]. Another drug, oxamniquine, which has demonstrated efficacy only against *S. mansoni*, is restricted to use in South America. Like praziquantel, treatment is administered orally via a single dose and has few reported side effects. In sum, the fact that commercially available drugs do not prevent reinfection, coupled with reports of drug resistance, making the search for alternative chemotherapeutic solutions to overcome current limitations urgent, whether through the development of new drugs or combination therapy involving PZQ [9].

In this context, natural products, which have been used in the treatment of human diseases for many years, are produced by living organisms, such as plants, animals, microbes, and marine organisms [18]. When derived from plants, natural substances are usually obtained from leaves, bark, stems, or roots, and may include alkaloids, phenolic compounds, flavonoids, terpenoids, tannins, saponins, and steroids [19]. Several diseases (such as malaria, dyspepsia, liver disorders, and glaucoma) are currently treated with drugs produced from plant-based bioactive entities, e.g., quinine, silymarin, artemisinin, or pilocarpine, among others [20]. Moreover, around 60% of the antiparasitic compounds used between 1981 and 2014 originated from natural products [18]. This study presents an overview of the anthelmintic activity exhibited by several natural products, including essential oils, in different in vitro and in vivo systems. In addition, we also describe useful tools for discovering new drugs that can enable reductions in both time and cost.

## 2. Plant-Derived Compounds

Despite the pharmaceutical industry’s growing interest in synthetic molecules for drug development, natural products remain a valuable source for new molecule discovery, including those with antiparasitic activity [21]. In this context, several reports have explored well-known natural products (e.g., quercetin, curcumin, pirplatine, and others), novel natural molecules [22,23,24] (Figure 1) and essential oils and their components [25] in the search for new antischistosomal drugs.

### 2.1. In Vitro Studies

Several in vitro studies (Table 1) have demonstrated the antischistosomal activity of a variety of natural molecules, especially against adult worms of *S. mansoni*. These reports mainly describe the ability of natural products to decrease worm motility and induce death via different pathways (Figure 2). The terpene nerolidol (3,7,11-trimethyl-1,6,10-dodecatrien-3-ol), also known as peruviol, was shown to promote a reduction in worm motility as well as death in adult parasites at concentrations ranging between 62.5 and 250 µM after 48 or 72 h of treatment; moreover, adult male parasites were found to be more susceptible to nerolidol than female worms [26]. In another investigation involving 10 triterpenes with the cucurbitane skeleton, balsaminol F and karavilagenin were identified as promising antischistosomal agents, since motor activity became significantly reduced (at 10–50 µM) and 100% death was observed in adult worms of *S. mansoni* at 100 µM with LC_50_ values of 14.7 and 28.9 µM respectively against 56-day-old adult *S. mansoni* [27]. In addition, licochalcone A, a characteristic chalcone of licorice, presented LC_50_ values of 9.12 and 9.52 µM against female and male adult worms, respectively, with impairment of motor activity observed at concentrations between 12.5 and 200 µM [28]. In fact, most of the compounds described in Table 1, especially those belonging to the terpene and chalcone classes, have been shown to reduce motility and induce death in adult *S. mansoni* worms [2,22,29,30,31,32,33,34,35,36,37,38,39].

Interestingly, the changes observed in motility and death of adult *S. mansoni* worms caused by the natural products described in Table 1 are mainly associated with alterations in the tegument [26,29,30,34,36,43,44,45,46,48]. The tegument is considered a key structure in the evasion of host immune response, acquiring nutrients, excreting catabolic products, and targeting drug absorption, among other physiological processes [49]. Therefore, it is an attractive target for the development of antischistosomal drugs. A natural product featuring the tegument of *S. mansoni* as an already characterized target is diterpene phytol. This natural product, at concentrations between 50 to 150 µg/mL, promotes severe tegument damage in schistosomes, such as body deformation, morphological disfiguring of the oral and ventral suckers, extensive sloughing, loss of tubercles, and shrinking [35,44]. Moreover, quantitative analysis revealed a concentration-dependent reduction in the number of intact tubercles after phytol treatment, with complete tubercle destruction observed at 100 µg/mL [44]. A reduction in the number of intact tubercles and morphological tegument alterations was also observed in adult *S. mansoni* worms treated with chalcones, especially licochalcone A and licoflavone B [33,36,48]. Transmission electron microscopy was used to visualize the formation of vacuoles of different sizes in the tegument and swelling in different regions of the integument due to the presence of sparse matrix after treatment with 10 µM of licochalcone A. In addition, licochalcone A also promoted swelling and degeneration of the mitochondria, as well as nuclear chromatin condensation, which were all correlated with increased superoxide anion levels and decreased superoxide dismutase activity. Interestingly, licochalcone A presented schistosomicidal activity without affecting the viability of mammalian cells (CHO-K1 cells; Chinese hamster ovary fibroblasts) at concentrations ≤400 µM [48].

Similarly, licoflavone B treatment caused massive disintegration of the tegumental surface in association with disruption of tubercles. These effects were accompanied by a pronounced inhibition of *S. mansoni* ATPase and ADPase activity, with resulting IC_50_ values of 23.78 and 31.51 µM, respectively, which was corroborated by docking studies involving licoflavone B and SmATPDase 1 [33]. The schistosomicidal activity of licoflavone B was observed at concentrations (25–200 µM) nontoxic to mammalian Vero cells [33]. Similar results were described by Pereira et al. [36], who used *S. mansoni* ATP diphosphohydrolases as a target and identified schistosomicidal activity in a series of chalcones without affecting cell viability [36].

Some natural products were also shown to interfere with the reproductive fitness of *S. mansoni*. Regarding oviposition, licoflavone B reduced the total number of eggs laid at sub-lethal concentrations (2.5, 5, and 10 µM) and inhibited 100% of egg-laying at 10 µM [33]. Oviposition was also affected by other natural products, such as dermaseptin 01, balsaminol F, karavilagenin C, phytol, dibenzylbutyrolactonic lignans, and curcumin [24,27,29,37,44].

Finally, some natural products demonstrated activity against *S. mansoni* cercariae (infective larval stage) as well as snails (Table 1) [24,32,47]. The alkaloid diethyl 4-phenyl-2,6-dimethyl-3,5-pyridinedicarboxylate exhibited potent cercaricidal activity (LC_100_ = 2 μg/mL) in addition to activity against adult *B. glabrata* (LC_90_ = 36.43 μg/mL) [33]. Barbatic acid, a lichen metabolite, provoked substantial molluscicidal activity against snails at concentrations ranging between 10.5 and 50 µg/mL, with optimal effects (100% lethality) observed at 25 µg/mL. In addition, barbatic acid also presented cercaricidal activity, completely eliminating cercariae at concentrations between 1 and 100 µg/mL after 60 min of drug exposure. Importantly, both cercaricidal and molluscicidal activity was found to occur at concentrations that did not alter *Artemia salina* viability [47]. Lastly, it is important to highlight the in vitro activity of curcumin. At different treatment times, LC_50_ values below 10 ug/mL demonstrated efficacy against cercariae and inhibited ability egg-laying capacity as well as egg hatchability, causing death in newborns, embryos, and adult *B. glabrata* snails [24].

### 2.2. In Vivo Studies

The antischistosomal effect of orally or intraperitoneally administered natural compounds has also been observed in several in vivo studies involving mice (Table 2). Figure 3 details the main outcomes identified in this literature review.

Initial work by Allam (2009) [50] demonstrated the anti-schistosomal activity of curcumin (400 mg/kg) in a murine model through the modulation of both cellular and humoral immune response. The intraperitoneal treatment of infected mice with curcumin resulted in a significant reduction in levels of interleukin (IL)-12 and tumor necrosis factor alpha (TNF-a) compared to the untreated infected group. This downregulation can be attributed to curcumin’s inhibition of nuclear factor kappa B (NF-κB), leading to a subsequent decrease in pro-inflammatory cytokines [68]. Moreover, the treated animals exhibited elevated levels of specific IgG and IgG1 antibodies against soluble worm and soluble egg antigens. The hepatoprotective property of curcumin has been noted in the context of liver fibrosis [51]. El-Agamy et al., 2011, [51] also reported similar antifibrotic effects in *S. mansoni* infection following oral curcumin treatment (300 mg/kg/day) for 2 weeks.

Interestingly, in mouse models, schistosome eggs or egg-derived antigens are considered potent inducers of a Th2-type immune response. A robust Th2 response has been shown to play a significant role in the development of granulomas around deposited eggs [57,69]. Numerous investigations have reported significant variations in histopathological findings between mice receiving treatment with natural products and those that did not, as evidenced by marked decreases in hepatic granulomas, fibrosis, and pro-inflammatory cytokines [53,55,66,67]. The recently evaluated compound plumbagin (5-hydroxy-3-methyl-1,4-naphthoquinone), derived from walnut trees, has been demonstrated to alleviate schistosome-induced hepatosplenomegaly, as well as to reduce hepatic granuloma and liver collagen content by 62.5% and 35.3%, respectively. Plumbagin exhibited immunomodulatory properties, as evidenced by increased levels of IL-10, while levels of IL-4, IL-13, IL-17, IL-37, IFN-γ, TGF-β, and TNF-α decreased [66]. Interestingly, a similar immunomodulatory profile was observed from another phenolic compound derived from the same source, Juglone (5-hydroxy-1,4-naphthoquinone) [67].

Studies conducted by El-Aal et al. [55] demonstrated that treatment with Paeoniflorin, a potential anti-schistosomal therapy, led to higher serum levels of TNF-α compared to healthy controls, infected controls, and Praziquantel-treated mice. The literature contains conflicting information regarding the complex role of TNF-α in schistosomiasis. While this cytokine can initiate apoptosis and provoke anti-fibrogenic effects, it has also been associated with granuloma formation and a fibrotic tissue development [50,55,56].

The oral administration of different natural compounds at doses ranging between 1.5 and 400 mg/kg demonstrated variable worm and egg burden reduction (31.8 to 100%) in infected mice [44,53,54,60,63,64]. In addition, Carvalho et al. [65] demonstrated the effective antischistosomal activity of asiaticoside (400 mg/kg) against *Schistosoma* spp. by inhibiting SmNTPDases—enzymes found in the worm tegument—which resulted in significantly reduced worm burden.

The findings in the reviewed studies indicate the natural compounds used for treatment via intraperitoneal route were all of the phenol class, or its derivatives [43,50,66,67]. For example, licochalcone A exhibits significant potential as a therapeutic drug. However, recent pharmacokinetic studies have indicated that its oral bioavailability is limited due to poor absorption and inactivation. Thus, alternative routes of administration, such as intraperitoneal or conjugation with nanoparticles, may be required to enhance therapeutic efficacy [62].

Several studies have shown female worms to be more susceptible to antischistosomal drugs than males [43,53,67]. In this scenario, Juglone has been proven to exert efficacious schistomicidal activity. Naphthoquinones react with the thiol groups of *S. mansoni* parasite proteins and inhibit enzymes essential for parasite survival [67]. Juglone was shown to significantly reduce the burden of both male and female worms by 63.12% and 52.1%, respectively. Notably, this compound demonstrated activity against both male and female worms, unlike other compounds that exhibit selective activity against one sex over another.

### 2.3. Essential Oils and Their Components in Use against Schistosoma Mansoni

Essential oils are characterized by a mixture of volatile and hydrophobic secondary metabolites. These oils constitute one of the principal fractions of chemical substances found in plants, presenting marked odors and being composed primarily of terpenoids and phenylpropanoids [70]. Essential oils have been evaluated for their anti-schistosomal potential at different stages of the parasite life cycles.

In a molluscicidal evaluation, *Eucalyptus* essential oils demonstrated bioactivity against *B. glabrata* eggs [71]. Essential oil extracted from *Eryngium triquetrum*, which contains aliphatic polyacetylene, also was found to be toxic to infected snails, exerting moderate effects on *B. glabrata* embryos in terms of inhibited egg hatching and snail development [72]. Among the compounds evaluated, the activity of terpene compounds found in some oils was described. The commercially available monoterpenes thymol and α-pinene demonstrated activity against *B. glabrata* snails, inducing mortality in a concentration-dependent manner, as well as inhibiting the enzymatic activity of acetylcholinesterase (AChE) extracted from snails [73]. In the context of biotechnology applications, nanoemulsions as a vehicle of the essential oil of *Xylopia ochrantha* (main compounds: bicyclogermacrene and germacrene D) caused between 50 to 100% mortality in *B. tenagophila*, *B. straminea*, and *B. glabrata* juveniles and adults after 48 h, and inhibited the development of eggs deposited by treated snails [74]. 

The chemoprophylactic action of *Pterodon pubescens* essential oil as an additive in different soap formulations was studied. Following the application of solutions containing different soap concentrations to the tails of mice, the animals were infected immediately, or 24 h later, with *S. mansoni* via caudal immersion. After 45 days of infection, different levels of protection were observed, ranging from 29 to 100% [75].

Cercaricidal activity was described for essential oil from the fresh aerial part of *Apium graveolens var. secalinum* (alpha- and beta-pinene, myrcene, limonene, cis-beta-ocimene, gamma-terpinene, cis-allo-ocimene, trans-farnesene, humulene, apiol, beta-selinene, senkyunolide, and neocnidilide) and in essential oils of *Eucalyptus* spp. (*E. cloeziana*, *E. deanei*, *E. exserta*, *E. maculata*, *E. punctate*, and *E. resinifera*) [71,76]. In studies using cedar oil, the authors attributed optimal cercaricidal activity to the penetration phase of cercariae, in which disruption of the cercarial glycocalyx alters the physiological processes related to osmoregulation, and may increase the absorption of toxic substances [77]. 

In in vitro studies involving *S. mansoni*, essential oil from the leaves of *B. dracunculifolia*, constituted mainly of oxygenated sesquiterpenes, such as (E)-nerolidol (33.51%) and spathulenol (16.24%), demonstrated high activity in a schistosomicidal assay, leading to the death of cultured pairs of adult worms [78]. Essential oil from *Plectranthus neochilus*, consisting of b-caryophyllene (1; 28.23%), a-thujene (2; 12.22%), a-pinene (3; 12.63%), b-pinene (4; 6.19%), germacrene D (5; 5.36%), and caryophyllene oxide (6; 5.37%), was considered active, but less effective than chemotherapy treatment in terms of worm pair separation, mortality, decreased motor activity, and tegumentary changes. However, this oil was associated with a dose-dependent reduction in both number and percentage of *S. mansoni* eggs [79]. Essential oil from *Ageratum conyzoides* L., whose main constituents are precocele I (74.30%) and (E)-caryophyllene (14.23%), was also considered less active than PZQ in the in vitro treatment of worms, with a dose-dependent reduction observed in the number of *S. mansoni* eggs [80]. Essential oil obtained from *Tetradenia riparia* leaves also reduced motility and decreased the percentage of developed eggs [81]. When *S. mansoni* worms were incubated with *Mentha* × *villosa* essential oil and its individual constituents (rotundifolone (70.96%), limonene (8.75%), transcaryophyllene (1.46%), and β-pinene (0.81%)), no anti-schistosomicidal activities were observed for transcaryophyllene or β-pinene. However, the use of this essential oil and rotundifolone or limonene did result in decreased adult worm motility and increased mortality [82].

In vitro assays involving *Baccharis trimera* demonstrated motility loss and death in *S. mansoni* within 30 h after exposure. Morphological alterations in the tegument were described in male worms, indicating desquamation on the surface of the tegument, as well as the destruction of tubercles and spines, resulting in smooth body surface areas. This essential oil also caused integumentary destruction in female worms, in addition to the destruction of the oral and acetabular cups [83]. Ultrastructural worm evaluation revealed bubble lesions, loss of tubercles in some regions of the ventral portion, tegument alterations, vacuoles in the region of the syncytial matrix, and glycogen granules near the muscle fibers [84]. Essential oil from *Foeniculum vulgare MILL*, whose main constituents are (E)-anethole (69.8%) and limonene (22.5%), exerted inhibitory effects on *S. mansoni* egg development, with less effective results than the positive control (PZQ) in terms of mated pair separation, mortality, and decreased motor activity [85]. Essential oils obtained from *Citrus limonia* leaves (Limonene-29.9%, β-pinene-12.0%, sabinense-9.0%, citronellal-9.0%, and citronellol-5.8%) and *C. reticulata* fruit peels (limonene-26.5%, γ-terpinene-17.2%, linalool-11.1%, octanal-8.0%, myrcene-6.2%, and capraldehyde-3.9%) exhibited moderate in vitro schistosomicidal activity against adult *S. mansoni* worms [86]. Essential oil from *Dysphania ambrosioides* (L.) (main constituents: cis-piperitone oxide monoterpenes-35.2%, p-cymene-14.5%, isoascaridol-14.1%, and a-terpinene-11.6%) showed in vitro parasite mortality in 100% of adult worm pairs after 24 h of treatment [87].

Although several in vitro studies have demonstrated the effects of different essential oils on *S. mansoni*, few in vivo studies have been conducted. Treatment with essential oil from fresh *Melaleuca armillaris* leaves (main constituents: 1,8-cineol-33.93%, terpinen-4-ol-18.79%, limonene-10, 37%, and B-pinene-6.59%) administered in mice twice a week for six weeks (150 mg/kg, orally), from the second week post-infection, significantly improved levels of glutathione and malondialdehyde and raised levels of vitamins C and E [88]. Matos-Rocha, 2020, [61] treated infected mice with *Mentha* × *villosa* essential oil (200 mg/kg) and rotundifolone (141.9 mg/kg) for five consecutive days, observing respective reductions of 72.44% and 74.48% in recovered *S. mansoni* after treatment.

## 3. Useful Tools for the Screening of New Drugs in Schistosomiasis

To reduce the time and costs associated with conventional methods of determining anti-*Schistosoma* activity in new drugs, which normally include laborious manual experiments based on phenotypic analysis by microscopy, several computational and experimental tools have been developed to enhance the search for new compounds [89,90,91]. These tools can not only aid in the identification of new substances with antischistosomal potential, whether of natural, synthetic, or semi-synthetic origin, but also aim to determine drug-likeness and guide proximal screening steps with regard to biological activity.

The approaches utilized for the selection and design of new compounds include structure-based drug design, in which a potential molecular target is characterized, as well as its participation in some metabolic event related to disease. In this type of approach, the use of three-dimensional (3D) molecular target structure, whether validated or putative, is essential in conducting protein–ligand interaction studies, as well as evaluating the forces involved in these interactions. This type of approach permits enhanced in silico virtual screening (VS) capability [92]. By contrast, drug design based on ligands does not necessarily depend on the structure of the potential target, but rather estimates parameters of the ligands themselves, such as structure, activity, and other important properties related to biological activity and drug-likeness. The most common methods used to perform these estimations are QSAR (quantitative structure–activity relationship) and QSPR (quantitative structure–property relationship) [93].

Computational methods employ algorithms and simulations that help predict physical–chemical characteristics and the potential for interaction between molecules through molecular docking, molecular dynamics, and molecular mechanics. Several studies have used these types of tools in the search for new compounds with activity against schistosomiasis, such as CADD (Computer-Aided Drug Design) and QSAR modeling. CADD enables the performance of virtual compound library screening and can simulate chemical modifications in compounds with already-known activity, aiming to improve physicochemical characteristics and enhance interaction with specific target ligands. Computational techniques for drug screening also include in silico analysis and methods for large-scale experimental data analysis, such as those evaluated by High Throughput Screening (HTS) [91].

In this context, databases of compound structures and molecular models of the main targets of the *S. mansoni* parasite have aided in the process of developing robust in silico tools that provide refined and reliable 3D structures. Moreover, genomic and protein databases play a very important role by providing sequences to obtain high-quality molecular models for investigation using in silico tools. The genomes of *S. mansoni*, as well as related species *S. japonicum* and *S. haematobium*, were first published in 2009, followed by updated versions [94,95], and can currently be freely accessed via the Wormbase ParaSite database (https://parasite.wormbase.org/; accessed on 15 May 2023). Recently, the availability of whole-genome sequencing and transcriptomic analysis has demonstrated the importance of a deeper understanding of this parasite’s gene expression. The integration of data can not only advance the development of novel schistosomiasis control strategies, especially regarding new drug discovery, but also aid in many diverse aspects of investigation, including variations in parasites that may reflect treatment response [96,97].

Concerning molecular targets, a variety of proteins have been identified as potential target molecules for the action of drugs and inhibitors. The protein data bank (PDB) (https://www.rcsb.org; accessed on 15 May 2023) is a repository containing experimentally determined 3D protein structures and protein models defined by computers, both of which are employed in CADD approaches. This database contains over 140,000 structures of *S. mansoni* proteins and protein domains, alone or with ligand complexes. A summary of the main potential drug targets of *Schistosoma* are presented in a study by Cheuka (2022) [98] along with information on function, method of identification/validation, examples of inhibitors or antagonists, and relevant phenotypic effects on schistosomes. Of these, some molecules are widely known and investigated, such as thioredoxin glutathione reductase (TGR), glutathione-S-transferase (GST), histone deacetylase (HDAC), and 20S proteasome, while others have been more recently identified and require further validation [98].

Currently, the use of artificial intelligence and machine learning have been integrated into tools that both seek to predict the diagnosis of disease [99] and the generation of three-dimensional models, in an attempt to reverse certain barriers, such as the need for crystallographic structures prior to the determination of reliable structural models. In this context, the launch of the AlphaFold tool [100] represents a revolution in obtaining structural models, as this prediction system employs artificial neural networks to accurately and rapidly predict 3D protein structures from primary amino acid sequences, without necessarily utilizing a previously developed model. Applications in schistosomiasis studies include obtaining new protein models, some of which can be found in the AlphaFold Protein Structure Database (AlphaFold DB) (https://www.alphafold.ebi.ac.uk/; accessed on 14 May 2023), which currently contains more than 15,000 structures for *S. mansoni*, including the main drug targets SmTGR and SmGST.

Several studies have used these tools to facilitate the choice of targets and/or optimal drug candidates, in addition to algorithms that promote a more efficient understanding of data produced on a large scale, as in HTS phenotypic analysis [91].

Through the application of CADD using the Molecular Operating Environment (MOE) tool to optimize compound structures and evaluate drug-likeness, 27 compounds and derivatives of African medicinal plants previously reported to present anti-*Schistosoma* effects in vitro or in vivo were evaluated. Computational analysis reduced screening candidates to just four molecules with potential activity on *Schistosoma* molecular targets, including TGR, GST, HDAC, and arginase [101].

The combination of structure- and ligand-based screening methods was also applied in the virtual screening of 1000 alkaloid structures isolated from plants of the *Menispermaceae* and *Apocynaceae* families to detect potential action against *S. mansoni*. This study employed QSAR approaches based on chemoinformatic tools available at the website Openmolecules.org, such as DataWarrior, to identify two alkaloids for use as a starting point for the development of new chemical compounds with anti-*Schistosoma* activity [102].

With regard to metabolic targets, QSAR-based virtual screening of *S. mansoni* thioredoxin glutathione reductase (SmTGR) inhibitors was performed using high content screening (HCS) in an effort to discover novel antischistosomal agents. QSAR models aimed at inhibiting SmTGR were applied to three subsets from the ChemBridge library (∼150,000 compounds), which selected 29 compounds for further testing via two HCS platforms based on image analysis of assay plates. Among these, 2-[2-(3-methyl-4-nitro-5-isoxazolyl)vinyl]pyridine and 2-(benzylsulfonyl)-1,3-benzothiazole, two compounds representing the new chemical scaffolds, were found to exert activity against schistosomula and adult worms at low micromolar concentrations, thus constituting promising antischistosomal agents [103]. A combination of computational techniques that included molecular docking studies also allowed for the evaluation of approximately 1000 insect-derived compounds with demonstrated inhibitory activity against SmTGR [103]. The applied approach included both the creation of a database of molecules derived from insects and the use of the PLIP (protein–ligand interaction profiler) tool for virtual screening and selection of the best candidates to conduct experimental testing.

Another example of combining CADD methods for the virtual screening of anti-*S. mansoni* drugs based on molecular docking is evidenced in a study by Moreira et al., in which a panel of 85,000 molecules from the Managed Chemical Compounds Collection (MCCC) of the University of Nottingham (UK) were investigated against five protein kinases (JNK, p38, ERK1, ERK2, and FES). Computational analysis narrowed the initial number of molecules down to 169, which were predicted to bind to SmERK1, SmERK2, SmFES, SmJNK, and/or Smp38, and were thus selected for testing in in vitro screening assays using schistosomula and adult worms. This combination of in silico and in vitro assays helped optimize the search for the most promising compounds, leading to a total of 89 molecules that were considered active by experimental assays [104], with 17 having suitable drug-likeness parameters. One of the experimental approaches aimed at detecting movement in adult worms was the WormAssay, a high-throughput screening motility assay that simultaneously performs parallel analysis on all wells of an entire plate [105]. WormAssay is another example of a useful, low-cost computer-aided tool that can be applied in the search for molecules against schistosomes.

In the search for inhibitors of venus kinase receptors (VKR), important to schistosome growth and egg deposition, 645 molecules from GlaxoSmithKline (GSK) set 2 were screened against one of the target proteins of *S. mansoni* (SmVKR2). This strategy combined the use of an initial in vitro experimental screening approach that applied the surface plasmon resonance (SPR) technique to determine molecule binding constants, with just 12 demonstrating molecular interactions on a micromolar level. These twelve molecules were then tested against *S. mansoni* ex vivo, resulting in the identification of four compounds with antiparasitic activity under testing including the WormAssay for phenotypic determination. Furthermore, the crystal structure of the kinase domain of SmVKR2 obtained by the authors improved in silico docking, thus paving the way to identify more potent inhibitors against the VKR2 receptor in the future [106].

Targeting potential epigenomic effects on the parasite, a structure-based virtual screening approach was applied in the search for histone deacetylase 8 (SmHDAC8) inhibitors using molecular docking of a compound library containing 550,000 molecules (the Interbioscreen database) against the crystallized structure of SmHDAC8 deposited in the PDB. The Glide docking program identified eight novel N-(2,5-dioxopyrrolidin-3-yl)-n-alkylhydroxamate derivatives, which were found to be active in the low micromolar range against smHDAC8 by utilizing an established in vitro assay for protein–ligand interaction and apoptosis induction [107].

## 4. Strategies Employing Functional Genomics Approaches and New Perspectives

Modern methods and technologies that combine functional genomic approaches, including genomics, transcriptomics, proteomics and epigenetics of *S. mansoni*, have shown the importance of achieving a deeper understanding of this parasite’s gene expression and the integration of data to advance the development of schistosomiasis control strategies. This is relevant not only with respect to the new drug discovery, but also in many diverse aspects, including populational variations that may reflect on the response to treatment, as well as mechanisms involved in the development of resistance to chemotherapy [96,97].

The genome of *S. mansoni*, as well as related species *S. japonicum* and *S. haematobium*, was first published in 2009, followed by updated revisions [94,95] which are freely available on the WormBase ParaSite database (https://parasite.wormbase.org/; accessed on 14 June 2023). Genomic data from publicly available databases serve as resources for establishing relationships between drugs and targets, allowing for the identification of compounds from searches based on sequence homology and functional motifs. Neves and colleagues demonstrated the use of this type of approach in their search for drug repositioning for schistosomiasis [102]. Whole-genome sequencing has also been applied with the aim of elucidating the impact of treatment on the parasite genome. For example, the action of PZQ was evaluated on natural populations of *S. mansoni* in areas where mass drug administration strategies had been conducted [108,109].

Data available from proteomics studies permit greater knowledge surrounding the profile of differentially expressed proteins under a variety of conditions to which the parasite is exposed. A comparative study of the excretory/secretory proteome of adult male and female worms identified approximately 1000 proteins, of which 370 and 140 were secreted solely or abundantly by males and females, respectively. The use of functional genomic analysis tools served to indicate which classes of proteins were more related to secretion by males than females, providing valuable information on host–parasite interplay and male–female interaction [110].

The response to treatment on the parasite’s proteome is also a relevant topic of investigation. For example, laboratory resistance selection of *S. mansoni* isolates through exposure to PZQ enabled the evaluation of impact on protein expression profile in worms exhibiting reduced sensitivity to PZQ in comparison to a susceptible population. This resulted in the identification of two proteins, Ca^2+^-ATPase and HSP70, that revealed differentiated expression under diverse analyzed conditions [111]. PZQ-resistant isolates were also compared regarding global changes in gene expression, with emphasis placed on determining the differential proteome between male and female worms. This approach enabled the identification of 60 differentially expressed proteins between exposed and non-exposed populations, with some proteins detected exclusively in females, whose susceptibility to PZQ was reduced in relation to male worms [112]. This type of approach greatly enhances the understanding of potential resistance mechanisms and also aids in establishing potential correlations with the drug’s mode of action.

In a similar context, many studies have also evaluated the profile of differentially expressed genes through RNA-seq, which has proven to be a very robust tool in identifying altered mRNA levels under both natural and experimental conditions [113,114]. More recently, transcriptomic single cell analysis has shown great potential in answering questions regarding parasite development, as well as its interaction with the environment and the host, unraveling possible heterogenous responses and transcriptomic dynamics [115,116].

Studies employing a combination of strategies, including bioinformatics tools, cheminformatics, and functional genomics, have been shown to efficiently select molecular targets involved in epigenetic regulatory processes, as reported by Padalino et al. [117], who utilized sequence homology, phylogenetic analysis, and a refined search for functional motifs, which, following experimental validation, permitted the reclassification of two components of the histone methylation machinery. In addition, posterior investigation of the potential of inhibition of these targets demonstrated the possibility of exploring the epigenetic pathway towards the development of next-generation drugs targeting schistosome epigenetic pathway components [117].

Thus, the use and integration of “omics” technologies has provided data deemed fundamental to defining new perspectives on the identification and application of natural products with anti-schistosomal properties.

## 5. Concluding Remarks

In conclusion, natural compounds have demonstrated significant potential as antischistosomal agents. Both in vitro and in vivo studies have evidenced the effectiveness of natural compounds against schistosomes under various approaches, including prophylactic interventions and adult parasite, cercariae, and schistosomula killing, as well as suppressive strategies that inhibit worm egg-laying. However, specific mechanisms of action, toxicological testing, and in vivo activities require better characterization in further investigations to allow transposing the use of natural products in clinical studies involving subjects with schistosomiasis. Overall, natural products have provided a valuable source of potential antischistosomal agents that may aid in the development of new treatments for schistosomiasis, a disease that continues to cause significant morbidity and mortality in endemic areas.

## Figures and Tables

**Figure 1 molecules-28-06807-f001:**
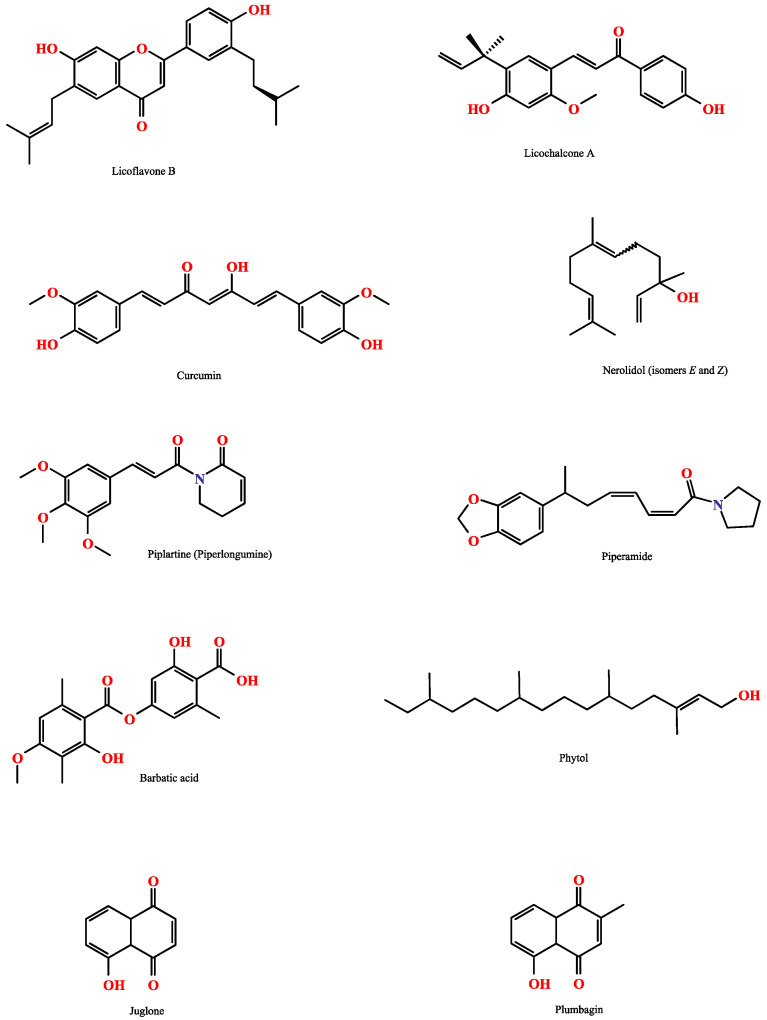
Chemical structures of natural products with antischistosomal activity demonstrated either in in vitro assays and/or murine models of infection.

**Figure 2 molecules-28-06807-f002:**
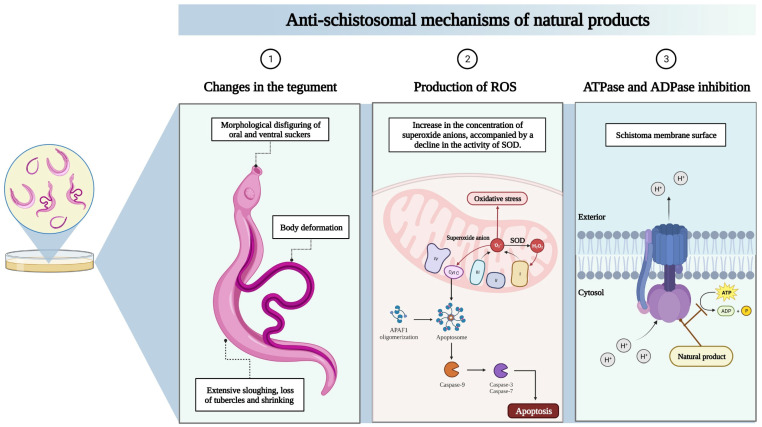
Main mechanisms of action of natural products against adult worms of *S. mansoni*. In general, natural products induced tegument damage in schistosomes associated with body deformation, morphological disfiguring of the oral and ventral suckers, extensive sloughing, loss of tubercles, and shrinking (1). In addition, some natural products, such as licochalcone A, promoted swelling and degeneration of mitochondria and nuclear chromatin condensation, which correlated with increased superoxide anion levels and decreased superoxide dismutase activity (2). Some natural products, mainly chalcones, inhibited *S. mansoni* ATPase and ADPase activity (3). The reactive oxygen species (ROS) are mainly produced at the electron transport chain (ETC) in the mitochondria, which are formed by transmembrane protein complexes (I–IV). During transportation, leaked electrons interact with oxygen to form superoxide anions (O_2_-) at complexes I and III. These complexes are the major source of superoxide and hydrogen peroxide (H_2_O_2_) since the O_2_- released can be reduced into H_2_O_2_ through a reaction catalyzed by superoxide dismutase (SOD) (Tirichen et al., 2021 [40]; Brand, 2016 [41]). The overproduction of superoxide anions leads to oxidative stress and activates transcription factors such as NF-κB and AP-1. Moreover, increased ROS in mitochondria can induce the release of transmembrane proteins such as cytochrome c, an electron carrier between complexes III and IV, into the cytosol that triggers the apoptotic machinery of the cell (Tirichen et al., 2021 [40]; Guerra-Castellano, 2018 [42]).

**Figure 3 molecules-28-06807-f003:**
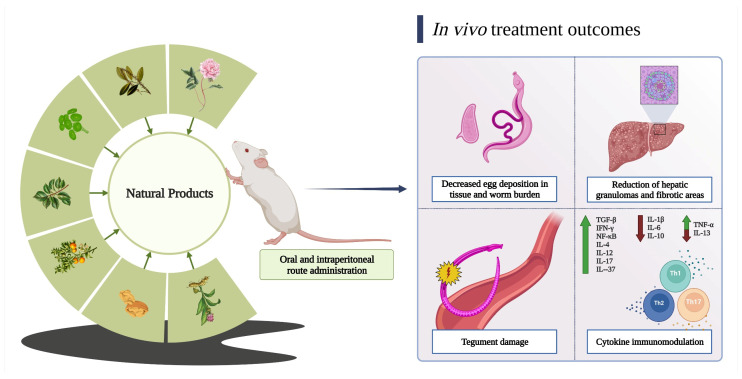
Main outcomes of in vivo treatments with natural products. Following intraperitoneal or oral administration, natural products have demonstrated an ability to reduce *Schistosoma* egg deposition in tissue and worm burden, leading to a reduction in the size of hepatic granulomas and fibrotic areas. Some compounds, such as nerolidol and piplartine, were observed to induce tegument damage as a mechanism of worm elimination. Moreover, plumbagin, curcumin, and other natural products exhibited different immunomodulatory properties by modulating cytokine production associated with Th1, Th2, and Th17 profiles.

**Table 1 molecules-28-06807-t001:** In vitro antischistosomal activity of compounds isolated from natural sources.

Molecules	Concentrations	Main Results	References
Dermaseptin 01	25, 50, 75, 100, 150, and 200 µg/mL	Dermaseptin 01 reduced motility and induced death in adult worms of *S. mansoni* at concentrations between 50 and 200 µg/mL. In addition, Dermaseptin 01 reduced the egg output of paired female worms and induced morphological alterations in the tegument of *S. mansoni*	[29]
Betulin, Oleanolic acid, Ursolic acid, Quercetin 3-*O*-β-d-rhamnoside, Quercetin 3-*O*-β-d-glucoside, Quercetin 3-*O*-β-d-glucopyranosyl-(1-2)- α-l-rhamnopyranoside, and Isorhamnetin 3-*O*-β-d-glucopyranosyl-(1-2)-α-l-rhamnopyranoside	50, 100, and 200 µM	Natural products reduced motor activity and caused death in adult *S. mansoni* worms	[22]
Pirplatine	7.5, 15, 30, and 60 µM	Piplartine treatment resulted in the death of all schistosomula in a concentration- and time-dependent manner. Microscopic observation revealed extensive tegumental destruction, including blebbing, granularity, and shortened *S. mansoni* schistosomula body length.	[23]
Balsaminol F and Karavilagenin C	10, 25, 50, and 100 µM	Balsaminol F and Karavilagenin presented LC_50_ values of 14.7 and 28.9 µM, respectively, against 56-day-old adult *S. manson*i. In addition, at 10–50 µM, both compounds caused significantly reduced worm motor activity and significantly decreased egg production. At 10–100 µM, both triterpenes separated adult worm pairs into males and females after 24 h	[27]
(+)-limonene epoxide	12.5, 25, 50, and 75 µg/mL	Treatment with compound reduced motility and induced death in adult *S. mansoni* worms at concentrations ≥25 µg/mL. Microscopic analysis revealed (+)-limonene epoxide mediated worm killing in association with tegumental destruction	[30]
Hesperidin	50, 100, and 200 µg/mL	Hesperidin, at 200 µg/mL, caused 100% mortality in 56-day-old adult worms within 72 h, with partial tegumental alterations observed in 10% of worms	[43]
N-[7-(30,40-methylenedioxyphenyl)-2(Z),4(Z)-heptadienoyl] pyrrolidine	10, 25, 50, and 100 µM	The isolated compound N-[7-(3′,4′-methylenedioxyphenyl)-2(Z),4(Z)-heptadienoyl] pyrrolidine promoted death of all adult worms of *S. mansoni* at 100 µM after 24 h of treatment	[31]
Phytol	12.5, 25, 50, 75, and 100 µg/mL	Treatment with phytol reduced worm motor activity and caused death. Confocal laser scanning microscopy analysis revealed extensive tegumental alterations in a concentration-dependent manner (50 to 100 µg/mL). Additionally, sublethal doses of phytol (25 µg/mL) reduced numbers of *Schistosoma mansoni* eggs	[44]
Diethyl 4-phenyl-2,6-dimethyl-3,5-pyridinedicarboxylate	1, 10, and 100 µg/mL	The alkaloid promoted the inhibition of movement and death in *S. mansoni* adult worms, accompanied by the formation of vesicles and vacuolization. In addition, the alkaloid exhibited a potent cercaricidal activity (LC_100_ = 2 μg/mL) as well as activity against adult snails (LC_90_ = 36.43 μg/ mL)	[32]
Nerolidol	15.6, 31.2, 62.5, 125, and 250 µM	Nerolidol reduced motor activity and caused death in adult *S. mansoni* worms. In addition, morphological alterations were observed in the tegument of worms (disintegration, sloughing, and surface erosion)	[26]
Licoflavone B	5, 10, 25, 50, and 100 µM	Licoflavone B (25 to 100 µM) caused 100% mortality, tegumental alterations, and reduced oviposition and motor activity in all adult worms, without affecting mammalian Vero cells. Licoflavone B also highly inhibited *S. manson*i ATPase (IC_50_ of 23.78 µM) and ADPase (IC_50_ of 31.50 µM) activity	[33]
Streptomycete-derived compound SF2446A2	0.5–10 µM	Treatment with 100 µM of SF2446A2 affected the gonads by impairing oogenesis and spermatogenesis. In addition, SF2446A2 caused disruptive effects on the tegument surface of *S. mansoni*	[45]
Cardol triene, Cardol diene, Anacardic acid triene, Cardol monoene, Anacardic acid diene, 2-methylcardol triene, and 2-methylcardol diene	12.5, 25, 50, 100, and 200 µM	Compounds Cardol diene and 2-methylcardol diene showed activity against *S. mansoni* adult worms, with LC_50_ values of 32.2 and 14.5 μM and selectivity indices of 6.1 and 21.2, respectively. Transmission electron microscopy revealed alterations in the tegument and mitochondrial membrane.	[34]
Phytol	25, 50, 75, 100, 125, and 150 µg/mL	Phytol reduced motility and induced death in adult *S. mansoni* worms at 150 μg/mL, with male worms more susceptible to treatment. On an ultrastructural level, phytol induced tegumental peeling, disintegration of tubercles and spines, as well as morphological disfiguring of oral and ventral suckers	[35]
Series of 38 terpenes	10, 20, 40, 80, 100, and 160 µM	Only dihydrocitronellol at 100 µM presented schistosomicidal activity after the maximal screening time of 120 h. Confocal laser scanning microscopy revealed severe tegumental damage induced by dihydrocitronellol in adult schistosomes	[46]
Barbatic acid	0.25, 0.5, 1, 10, 25, and 100 µg/mL	Barbatic acid exhibited molluscicidal activity against snails, especially at 25 µg/mL, with 100% lethality. In addition, barbatic acid presented cercaricidal activity, completely eliminating cercariae at concentrations between 1 and 100 µg/mL	[47]
Terrein, Butyrolactone I, and butyrolactone V	25–1297.3 µM	All compounds reduced motility and induced death in adult *S. mansoni* worms at concentrations between 235.6 and 454.1 µM	[28]
Licochalcone A	3.125, 6,25, 12,5, 25, 50, 100, and 200 µM	Licochalcone A reduced the number of *S. mansoni* eggs and affected egg development in adult worms. Drastic changes in the tegument of *S. mansoni* adult worms and alterations in mitochondria and chromatin condensation were related to increased superoxide anion levels and decreased superoxide dismutase activity in adult *S. mansoni* worms	[48]
A series of 15 chalcones	10, 50, and 100 µM	Chalcones, especially 1 and 3, induced adult worm death, reduced motility, and caused changes in the tegument of adult *S. mansoni* worms	[36]
(-) Hinoquinin, (-)-Cubebin, Yatein, 5-Methoxyyatein, Dihydrocubebin, and Dihydroclusin.	10, 25, 50, and 100 µM	(-) Hinoquinin, (-)-Cubebin, Yatein, and 5-Methoxyyatein decreased motor activity in adult *S. mansoni* worms. All compounds, except Dihydrocubebin, were found to separate adult worm pairs and reduce egg numbers after 24 h of treatment	[37]
Curcumin	1.56, 3.125, 6.25, 12.5, 25, 50, and 100 µg/mL	Curcumin presented LC_50_ values <10 µg/mL against cercariae. Treatment with curcumin affected egg-laying capacity and egg hatchability, causing death in newborns, embryos, and adult *B. globrata* snails.	[24]
6-[8(Z)-pentadecenyl] anacardic, 6-[10(Z)-heptadecenyl] anacardic acid, and 3-[7(Z)-pentadecenyl] phenol	1, 10, and 100 µM	All compounds presented activity against *S. mansoni*, killing 100% of adult *S. mansoni* worms at 100 µM	[38]
Anemonin	1 and 10 µM	Anemonin demonstrated activity against adult *S. mansoni* and newly transformed schistosomules (49% activity against adult *S. mansoni* at 10 µM and 41% activity against newly transformed schistosomules at 1 µM)	[39]

ATPase, adenosine triphosphatases; LC_50_, lethal concentration of 50%; LC_90_, lethal concentration of 90%; LC_100_, lethal concentration of 100%.

**Table 2 molecules-28-06807-t002:** In vivo antischistosomal activity of compounds isolated from natural sources.

Molecules	Route	Dose	Main Results	References
Curcumin	Intraperitoneal	400 mg/kg/day	Curcumin reduced worm and tissue egg burden, hepatic granuloma volume, and liver collagen content by 44.4%, 30.9%, 79%, and 38.6%, respectively	[50]
Curcumin	Oral	300 mg/kg/day	Curcumin treatment exerted antifibrotic effects in *S. mansoni*-infected mice	[51]
Phytol	Oral	40 mg/kg/day	A single dose of phytol (40 mg/kg) resulted in total and female worm burden reductions of 51.2% and 70.3%, respectively. Also, reduced numbers of eggs were found in feces (76.6%), with a lower frequency of immature eggs	[44]
Hesperidin	Intraperitoneal	100 mg/kg/day	Reductions of 50, 45.2, 50, and 47.5% in males, females, worm pairs, and total worm burden, respectively. In addition, respective reductions, based on the number of eggs/g of tissue, of 41.5, 63.7, and 58.6% were observed in the liver, intestine, and liver/intestinal tissue combined	[43]
Triphenylphosphonium	Oral	400 mg/kg/day	Triphenylphosphonium salts 10 and 11 resulted in low worm burden reductions against *S. mansoni* of 21.9% and 22.2%, respectively. Both compounds were well-tolerated by mice	[52]
Epiisopiloturine	Oral	40, 100, and 300 mg/Kg/day	Treatment with epiisopiloturine at 40 mg/kg reduced total worm burden by 50.2%, as well as hepatosplenomegaly, egg burden in feces, and granuloma diameter. Electron microscopy revealed a loss of important features in the parasite tegument	[53]
Nerolidol	Oral	100, 200, and 400 mg/kg/day	Nerolidol (100, 200, or 400 mg/kg) reduced worm burden and egg production in mice infected with adult schistosomes. Treatment with the highest concentration reduced total worms by 70.06% and immature eggs by 84.6%. Microscopic observations revealed that nerolidol-mediated worm killing was associated with tegumental damage	[54]
Paeoniflorin	Oral	50 mg/kg/day	Paeoniflorin treatment decreased worm burden, as well as immature and mature eggs, with reductions in hepatic granuloma size and fibrotic areas	[55]
7-epiclusianone	Oral	100 or 300 mg/kg/day	7-epiclusianone showed significant schistosomicidal in vivo activity following treatment with 300 mg/kg for 5 days	[56]
Allicin	Oral	0.5 μM/mouse	Prophylactic administration of allicin in infected mice significantly reduced worm burden. Serum concentrations of liver fibrosis markers and proinflammatory cytokines were also reduced	[57]
Series of 15 chalcones	Oral	400 mg/Kg/day	Chalcones 1 and 3 demonstrated moderate schistosomicidal activity with total worm burden significantly reduced by 32.8% and 31.8%, respectively, at a single oral dose (400 mg/kg)	[36]
Epiisopilosine alkaloid	Oral	100 or 400 mg/Kg/day	A single dose of epiisopilosine significantly decreased total worm load by 57.78 and 60.61% at doses of 400 and 100 mg/Kg, respectively. In addition, epiisopilosine significantly reduced eggs number and decreased hepatosplenomegaly	[58]
Piplartine	Oral	100, 200 or 400 mg/kg/day	Treatment with the highest piplartine dose (400 mg/kg) caused a significant (60.4%) reduction in total worm burden in mice harboring adult parasites. Microscopy revealed substantial tegumental alterations in parasites recovered from mice	[59]
Gomphoside monoacetate and Uscharin	Oral	10 mg/kg/day	Only gomphoside monoacetate (10 mg/kg) demonstrated activity against *S. mansoni*, with a low worm burden reduction of 38%	[60]
Rotundifolone	Oral	35.9, 70.9 and 141.9 mg/Kg/day	Rotundifolone (141.9 mg/kg) significantly reduced fluke burden by 74.48%. Marked reductions in liver, intestinal, and fecal fluke burden, together with changes in the oogram pattern were observed. Treatment affected the viability of both mature and immature eggs	[61]
Licochalcone A	Oral; intraperitoneal	1.5 or 2.5 mg/kg/day (oral); 25 mg/kg/day (intraperitoneal)	Oral treatment with L-SLNs decreased worm burden. However, under intraperitoneal administration, both free licochalcone A and L-SLNs significantly decreased worm burden and intestinal egg load	[62]
Carvacryl acetate	Oral	100, 200, or 400 mg/kg/day	Carvacryl acetate (400 mg/kg) showed moderate efficacy against *S. mansoni*, with slightly reduced worm burden (32–40%). Egg production was markedly reduced (70–80%)	[63]
Cardamonin	Oral	400 mg/kg/day	Oral treatment with cardamonin (400 mg/kg) demonstrated efficacy against *S. mansoni*, with decreased total worm load in 46.8% of mice and a 54.5% reduction in egg numbers	[64]
Asiaticoside	Oral	400 mg/kg/day	A single oral dose (400 mg/kg) of asiaticoside presented significant in vivo antischistosomal efficacy, markedly decreasing total worm and egg burden	[65]
Plumbagin	Intraperitoneal	20 mg/kg/day	Mice treated with plumbagin (20 mg/kg) showed reductions of 64.28% and 59.88% in male and female worms, respectively. Plumbagin treatment also alleviated schistosome-induced hepatosplenomegaly and reduced hepatic granuloma and liver collagen content	[66]
Juglone	Intraperitoneal	2 mg/Kg/day	Treatment with the compound reduced male and female worms by 63.1% and 52.1%, respectively. The number of eggs/g of tissue in the liver and intestine were also reduced. Juglone decreased hepatic granuloma size and collagen fiber deposition. Mice treated with juglone presented significantly lower levels of IL-4, IL-13, IL-37, TNF-α, TGF-β, and IFN-γ than PZQ mice	[67]

L-SLNs, LicoA-loaded solid lipid nanoparticles.

## Data Availability

Not applicable.

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
