# Peer review of "Therapeutic Potential of Natural Products in the Treatment of Schistosomiasis"

_molecules, 2023, doi:10.3390/molecules28196807_

Round 1

Reviewer 1 Report

Dear Authors,

 I have carefully reviewed your manuscript titled " Therapeutic Potential of Natural Products in Treatment of Schistosomiasis " and appreciate the effort put into this work. Overall, it is a well-conducted review that provides valuable insights into the utilization of natural products for the treatment of schistosomiasis. However, I would like to address a few points that would enhance the quality and impact of your study.

Firstly, while the review effectively summarizes the existing knowledge in the field, it appears to lack new insights and perspectives, as acknowledged by the authors themselves in section 3. In today's scientific landscape, we have access to an abundance of genomic, transcriptomic, proteomic, and epigenetic data, which offer an opportunity to delve deeper into the analysis and propose natural products as a valuable tool for the treatment and control of schistosomiasis. Therefore, I recommend dedicating a new section to discuss and cite modern approaches utilizing functional genomic techniques and natural products. This inclusion would greatly strengthen the manuscript and provide readers with a more comprehensive understanding of the topic.

Furthermore, considering the significant number of molecules that have already been studied, as greatly highlighted in your manuscript, it would be valuable to expand upon the perspectives presented in your review and explore realistic treatment applications beyond the confines of wet-lab studies. To achieve this, I encourage you to contemplate what is currently missing or required to progress further in this field. Addressing this aspect would greatly contribute to the practical implementation of natural products for the treatment of schistosomiasis.

In conclusion, I commend your efforts in compiling a comprehensive review of the use of natural products in schistosomiasis treatment. To augment the impact of your work, I suggest incorporating a section dedicated to modern approaches utilizing functional genomics and expanding on realistic treatment applications based on the existing body of knowledge. These revisions will not only strengthen the manuscript but also provide valuable insights for researchers and practitioners working in the field of schistosomiasis.

Thank you for considering my suggestions, and I look forward to seeing an updated version of your manuscript.

 General remarks:

Line 39: Infection occurs when the individual comes into contact with water containing cercariae. These are released by intermediate host snails, which may be of the genus Biomphalaria (for S. mansoni), Oncomelania (for S. japonicum) or Bulinus (for S. haematobium). Cercariae can survive in water for 1 to 3 days, but their ability to infect rapidly reduces within a few hours. Add appropriate reference

Line 67: The pathology of schistosomiasis is closely related to the large amount of eggs de-67 posited by females and these accumulate in the liver/intestine or bladder/urogenital sys-68 tem [2,9].. Could other factors be related to the pathology of schistosomiasis? Host genomic background or parasite virulence?

Line 69: The disease has 3 phases: acute (occurs after primary infection, being more com-69 mon in travelers or immigrants to endemic areas), stabilized, and late chronic (commonly observed in individuals living in endemic areas [2]. Please describe the differences between each phase.

Author Response

Reviewer number: 1

Reviewer's remarks: I have carefully reviewed your manuscript titled " Therapeutic Potential of Natural Products in Treatment of Schistosomiasis " and appreciate the effort put into this work. Overall, it is a well-conducted review that provides valuable insights into the utilization of natural products for the treatment of schistosomiasis. However, I would like to address a few points that would enhance the quality and impact of your study.

Author´s response: We appreciate the reviewer for revising our manuscript and for the helpful comments. Below, we listed the answers to each question.

Reviewer's remarks: Firstly, while the review effectively summarizes the existing knowledge in the field, it appears to lack new insights and perspectives, as acknowledged by the authors themselves in section 3. In today's scientific landscape, we have access to an abundance of genomic, transcriptomic, proteomic, and epigenetic data, which offer an opportunity to delve deeper into the analysis and propose natural products as a valuable tool for the treatment and control of schistosomiasis. Therefore, I recommend dedicating a new section to discuss and cite modern approaches utilizing functional genomic techniques and natural products. This inclusion would greatly strengthen the manuscript and provide readers with a more comprehensive understanding of the topic.

Author´s response: We agree with the reviewer. A new subtopic was incorporated in topic 3, discussing more modern approaches as suggested by the reviewer.

Reviewer's remarks: Furthermore, considering the significant number of molecules that have already been studied, as greatly highlighted in your manuscript, it would be valuable to expand upon the perspectives presented in your review and explore realistic treatment applications beyond the confines of wet-lab studies. To achieve this, I encourage you to contemplate what is currently missing or required to progress further in this field. Addressing this aspect would greatly contribute to the practical implementation of natural products for the treatment of schistosomiasis.

Author´s response: The information requested by the reviewer was added in the final considerations of the article.

Reviewer's remarks: In conclusion, I commend your efforts in compiling a comprehensive review of the use of natural products in schistosomiasis treatment. To augment the impact of your work, I suggest incorporating a section dedicated to modern approaches utilizing functional genomics and expanding on realistic treatment applications based on the existing body of knowledge. These revisions will not only strengthen the manuscript but also provide valuable insights for researchers and practitioners working in the field of schistosomiasis.

 Author´s response: Thanks to the reviewer for all constructive comments. We believe that the main points highlighted by the reviewer were improved and the scientific quality of our work increased.

Reviewer's remarks: Line 39: Infection occurs when the individual comes into contact with water containing cercariae. These are released by intermediate host snails, which may be of the genus Biomphalaria (for S. mansoni), Oncomelania (for S. japonicum) or Bulinus (for S. haematobium). Cercariae can survive in water for 1 to 3 days, but their ability to infect rapidly reduces within a few hours. Add appropriate reference

 Author´s response: An appropriate reference was added to this paragraph.

Reviewer's remarks: Line 67: The pathology of schistosomiasis is closely related to the large amount of eggs de-67 posited by females and these accumulate in the liver/intestine or bladder/urogenital sys-68 tem [2,9].. Could other factors be related to the pathology of schistosomiasis? Host genomic background or parasite virulence?

Author´s response: Based on the reviewer's comment, a better discussion of factors related to pathology was performed in the Introduction section.

Reviewer's remarks: Line 69: The disease has 3 phases: acute (occurs after primary infection, being more com-69 mon in travelers or immigrants to endemic areas), stabilized, and late chronic (commonly observed in individuals living in endemic areas [2]. Please describe the differences between each phase.

 Author´s response: An appropriate paragraph was added in the Introduction section explain differences between each phase.

Reviewer 2 Report

1. At least a paragraph on the importance of natural products in the treatment of disease should be included in the introduction part.

2. The structure of important constituents reported within the manuscript should be drawn and represented as Figures. e.g. line 109-110 active constituents (a). "terpene nerolidol (3,7,11-trimethyl-1,6,10-109 dodecatrien-3-ol), also known as peruviol"; (b). balsaminol F and karavilagenin; as well as all the active constituents given in Table 1.

3. The reference column in Table 1 and Table 2 should place in the last column.

4. I think substantial improvements are required to make the manuscript more informative and attractive. The manuscript looks monotonous and need coluored figures to make them more appealing.

Minor editing of English language required

Author Response

Reviewer number: 2

Reviewer's remarks: 1. At least a paragraph on the importance of natural products in the treatment of disease should be included in the introduction part.

Author´s response: An appropriate paragraph was added in the Introduction section as requested by reviewer.

Reviewer's remarks: 2. The structure of important constituents reported within the manuscript should be drawn and represented as Figures. e.g. line 109-110 active constituents (a). "terpene nerolidol (3,7,11-trimethyl-1,6,10-109 dodecatrien-3-ol), also known as peruviol"; (b). balsaminol F and karavilagenin; as well as all the active constituents given in Table 1.

Author´s response: Aiming to increase the scientific quality of our work and meet the reviewer's request, figure 1 containing the chemical structure of the most active natural products was added in the body of the work.

Reviewer's remarks: The reference column in Table 1 and Table 2 should place in the last column.

Author´s response: Modified as suggested by reviewer.

Reviewer's remarks: I think substantial improvements are required to make the manuscript more informative and attractive. The manuscript looks monotonous and need coluored figures to make them more appealing.

Author´s response: Based on the reviewer's comment, three figures have been added to the body of the paper.

Reviewer 3 Report

attached file

Requires minor clarification - detialed in attached file.

Author Response

Reviewer number: 3

Reviewer's remarks: This review article describes naturally occurring compounds that exhibit activity against Schistosoma spp. The therapeutic use of these compounds requires further investigation and potential development for use in Schistosomiasis chemotherapy as this currently relies largely on a single anthelmintic (Prazequantel). The review is wide ranging, encompasses many bioactive compounds, and closes by discussing the application of bioinformatic analysis and SAR to help identify candidate molecules for further investigation. The current paper is perhaps more extensive than another recent review [F. L. Mtemeli, J. Ndlovu, G. Mugumbate, T. Makwikwi & R. Shoko (2022); Advances in schistosomiasis drug discovery based on natural products., All Life, 15:1, 608-623, DOI: 10.1080/26895293.2022.2080281 (not cited)] however lacks the clarity and focus of the earlier article.

Author´s response: We appreciate the reviewer for revising our manuscript and for the helpful comments. Below, we listed the answers to each question. We emphasize that the suggestions made by the reviewers were fully met and we believe that our review now has adequate quality and clarity for Molecules readers.

Reviewer's remarks: There are also numerous spelling errors throughout that need to be corrected and some inaccuracies. The manuscript therefore requires major revision prior to publication. See some specific comments below:

  • Line 17; ‘the biodiversity’ – incorrect terminology.
  • Line 21; ‘that have demonstrated to be’ – correct English.
  • Line 31; correct spelling of haematobium’.
  • Line 33; correct spelling of ‘haematobium’ and ‘mansoni’.
  • Line 35; correct spelling of ‘Philippines’.
  • Line 36; correct spelling of ‘guineensis’.
  • Line 38; correct spelling of ‘Eastern’.
  • Lines 43-44; Schistosomula develop quickly after entering host and before reaching the lungs?
  • Line 50; correct spelling ‘haematobium’.
  • Line 57; unclear. The water is contaminated with eggs/microcidia from human feces. Reword sentence.
  • Lines 67 – 71; Short paragraph describing the ‘Pathology’ of the infection. Seems very rushed. Does it enhance the review?
  • Line 79-80; Route of Praziquantel administration? • Line 81; correct chemical abbreviation ‘Na+ ’. • Lines 84-89; clarify this sentence.
  • Lines 11-112; clarify ‘…in concentrations ….female worms’.
  • Line 114; ‘promising’ • Lines 113-117; clarify. Balsamino - LC50 14.7uM / karavilagenin 28.9uM after 24 hour incubation.
  • Line 128; correct spelling ‘tegument’.
  • Line 132; correct spelling '’tegument'.
  • Line 141-144; clarify and correct English.
  • Line 147; correct spelling '’tegumental'.
  • Line 157 & 165; avoid using 1st person (we).
  • Line 158; missing close bracket.
  • Line 170-171; Clarify that the studies use mouse models.
  • Line 175; Reference 43 looks at Schistosomes – not nematodes – they are trematode change to ‘anti-schistosomal’.
  • Lines 175-181; Clarify. Break into shorter sentences. What does ‘low levels of IL-12’ mean – low compared to what? Give SWAP and SEA in full.
  • Line 182-183; This is a direct copy from ref 43. Please paraphrase to avoid plagiarism.
  • Line 188-189; Ref 58 looks at Juglone (5-hydroxy-1,4-naphthoquinone). Plumbagin is 5-hydroxy-3-methyl-1,4-naphthoquinone).
  • Lines 191-193; Inaccurate. Some cytokines were found to increase in concentration (IL-10 /IL-17) while others fell (IL-4, IL-13, IL-37, TNF-A, TGF-B and IFN-G. Further, not all of these cytokines are pro-inflammatory.
  • Line 203; ‘Prospective’ – wrong word choice – multiple references are cited.
  • Line 220; Think you mean ‘essential oils’.
  • Line 223; ‘schistosomiasis potential – think you mean ‘anti-schistosomal’.
  • Line 233-234; clarify ‘showed bioactivity in egg masses snails’.
  • Line 234; correct ‘B. glabrata’.
  • Line 246; clarify end of sentence ‘and rendered unfeasible deposited eggs by these snails’. • Line 250; delete ‘of S. mansoni’.
  • Line 269; do not use contractions - ‘didn’t’.
  • Line 286; delete ‘in the experimental schistosomiasis’.
  • Line 325-328; clarify sentence. • Lines 337-338; change to ‘’be accessed freely in the Wormbase…’.
  • Line 345-346; delete ‘and pointed out’.
  • Line 356; delete ‘and’ change ‘still needs more’ to ‘require further’. • Line 421; correct English ‘These 12….against’.

Author´s response: All errors and inaccuracies flagged by the reviewer were corrected and the article was reviewed by a native English speaker.

Reviewer's remarks: Tables 1 and 2. These contain large amounts of useful information but the presentation requires further thought to allow the data to be easily accessible. Consider adding additional columns eg ‘concentration’ ‘life cycle stage’ ‘ class of molecule’ ‘source of molecule’ ‘parasite species’ ‘administration route’ ‘treatment regimen’.Consider changing to landscape format General.

Author´s response: The format of tables for landscape has changed. In addition, information on concentration, route of administration, and dose has been added in the new version of the manuscript.

Reviewer's remarks: Manuscript reads like a list with many paragraphs comprising one or two long complex sentences.

Author´s response: Adjusted as requested by the reviewer.

Reviewer's remarks: Mechanism of action of some of the compounds could be better described.

Author´s response: The mechanism of action of some compounds was better explored in section 2.

Reviewer's remarks: Are any of the compounds used therapeutically in other infections?

Author´s response: Many of the mentioned natural products have already been tested and validated for other parasitic diseases, cancer and inflammatory diseases. However, despite the promising preclinical evidence, none of the mentioned molecules is among the drugs available in the clinic.

Reviewer's remarks: Section 2.3 jumps between intermediate host and parasite, need to present logically.

Author´s response: Section 2.3 has been reorganized to make the logical writing sequence more appropriate. Thanks for the signage.

 Reviewer's remarks: No mention of any effects / toxicity of the natural compounds on the mammalian host (human)

Author´s response: Section 2 was restructured to contain information about the cytotoxicity/toxicity of the molecules under evaluation.

Reviewer's remarks: Although the list is extensive some potential therapeutics appear to have been missed – eg Zingiber, Pterocarpus – together with their respective references.

Author´s response: The focus of our work was a survey of pure molecules and essential oils obtained from plant sources. The natural products mentioned by the reviewer are described in the literature as extracts with a good therapeutic profile, and they do not fall within the scope of our work.

Round 2

Reviewer 2 Report

Manuscript is ok

Author Response

We thank the reviewer for all the constructive comments in the review process that were fundamental to achieve the quality of the current version of the work.

Reviewer 3 Report

Most comments have been addressed.

Line 182-183 (now 303-304); Although this is only a short sdentence it is a direct copy from ref 43 (now 48; Allam et al 2009). Please paraphrase and reference to avoid plagiarism.

Inclusion of the additional material enhances the manuscript.

There are still some grammatical improvements required but these should be picked up during the editing process.

Author Response

Reviewer's remarks: Line 182-183 (now 303-304); Although this is only a short sentence it is a direct copy from ref 43 (now 48; Allam et al 2009). Please paraphrase and reference to avoid plagiarism.

Author´s response: Modified as suggested by reviewer.

Reviewer's remarks: There are still some grammatical improvements required but these should be picked up during the editing process

Author´s response: The entire text has been proofread by a Native American as suggested.
